# Designing Dementia Care Pathways to Transform Non Dementia-Friendly Hospitals: Scoping Review

**DOI:** 10.3390/ijerph18179296

**Published:** 2021-09-03

**Authors:** Jorge Riquelme-Galindo, Manuel Lillo-Crespo

**Affiliations:** 1Faculty of Health Sciences, University of Alicante, 03690 Alicante, Spain; manuel.lillo@ua.es; 2HLA Vistahermosa Hospital, 03015 Alicante, Spain

**Keywords:** built environment, dementia, Alzheimer’s disease and associated disorders, clinical pathway, environment design, care pathway

## Abstract

People with dementia (PwD) occupy around 25% of the hospital beds. Once PwD are admitted to hospitals, their cognitive impairment is not considered in most of the cases. Thus, it causes an impact on the development of the disease becoming a stressful situation as care plans are not adapted to PwD. The aim of this study was to explore the published core elements when designing a dementia care pathway for hospital settings. A scoping review was conducted to provide an overview of the available research evidence and identify the knowledge gaps regarding the topic. This review highlights person-centered care, compassionate care and end-of-life process as some of the key elements that should integrate the framework when designing a dementia care pathway. Architectonical outdoor and indoor hospital elements have also been found to be considered when adapting the healthcare context to PwD. Findings provide information about the key points to focus on to successfully design dementia interventions in hospital environments within available resources, mostly in those contexts in which national dementia plans are in its infancy. Hospitals should transform their patients’ routes and processes considering the increasing demographic changes of people with cognitive impairment.

## 1. Introduction

### 1.1. Background 

Dementia population data predictions will continue to increase in the next decades. The International Alzheimer’s Association estimates that 75.6 million people will live with dementia in 2030, reaching 135.5 million in 2050 [1]. The world economic impact of neurodegenerative diseases in 2020 was estimated in more than 1.2 trillion US dollars corresponding to the 80% of such investments to Europe and North America [2]. Alzheimer’s disease and other dementias represent the highest percentage of neurodegenerative diseases [3]. Consistent with this, several healthcare systems worldwide have implemented specific Dementia National Plans [4] for years, while countries such as Spain, among other European nations, are still in its early stages with its first Alzheimer’s and Dementia Plan for the period 2019–2023 [5].

The number of people with dementia (PwD) being admitted in emergency services continues rising. In countries, such as England, the number of admitted PwD has increased more than 35% in five years [6]. Furthermore, PwD are three-fold higher than those without a diagnosis of any type of dementia of contracting COVID-19, with a higher risk associated to develop a severe form of COVID-19 requiring hospitalization [7]. Therefore, the need of transforming health environments into dementia-oriented settings remains urgent. Governments at worldwide level have expressed their commitment to share systematic approaches to prevent, diagnose and treat dementia and are developing plans to achieve specific goals to create dementia-friendly environments in healthcare facilities and outdoors environments [8]. 

### 1.2. Dementia Care Pathways 

Care pathways (CP) are implemented by adapting clinical practice guidelines or protocols to the actual and specific functioning in one health institution [9], in order to provide the sequence of actions necessary to achieve objectives efficiently reducing variability in clinical practice [10,11]. CPs are developed by multidisciplinary teams, therefore guiding professionals to better understand their roles and responsibilities in improving the level of care provided [12]. 

Hospitalization has shown to be a challenging time for those whose dementia disease makes it difficult for them to adapt to an unfamiliar environment and unknown people, such as healthcare staff [13]. This situation contributes in making PwD have almost double hospital stays per year than any other group of older people [14]. In terms of economic analysis, cost-related dementia hospitalization is estimated to be almost three times higher compared with non-dementia patients [15]. Thus, evidence has shown that implementation of clinical pathways (CP) in hospitals decreases the length of stay as well as other avoidable medical costs [16,17]. However, most of the literature about dementia care pathways has been studied and piloted in those 15% worldwide countries that account for a concise dementia national plan [18]. These national plans are of high importance as compared with international or any other initiative. These are culturally and demographically the most specific strategies to promote public awareness of dementia, improving the quality of health care, social care and support for PwD and their families. Table 1 shows the list of countries that account for a dementia national plan. 

### 1.3. Dementia-Friendly Interventions

Dementia-friendly environment interventions have helped to design adapted environments for PwD considering their experiences, common behaviors, habits and enablers to facilitate their daily routines [19]. These interventions have contributed to tackle the belief of considering health organizations as unknown and not friendly environments. Different initiatives have been successfully implemented in North European countries for being inclusive with this population, where laws, policies and strategies to support PwD are stablished [20,21]. In these regions, dementia-friendly hospitals and healthcare centres can be found as an architectonical and cultural adaptation to PwD [22].

Findings from studies conducted in those dementia culturally friendly contexts cannot be always generalized to all countries, as the healthcare systems are likely to be different in terms of healthcare staff training and ratios per patient, work practices, social and economic administrations, infrastructure, legislations and regulations and the culture of each country. 

Thus, the present scoping review aimed to determine the scope and volume of available research conducted about care pathways and hospital friendly interventions for people with dementia that can be implemented in those contexts and settings in which these interventions at national level are still in their early stages.

## 2. Materials and Methods

The scoping review was the type of literature search selected by following the PRISMA Extension for Scoping Reviews-PRISMA-ScR [23] and the Joanna Briggs Institute guidance [24], with the purpose to identify relevant publications, both scientific and grey literature, regarding the key concepts of the topic explored with direct implications into healthcare settings. This review aims to explore how the design of specific interventions for PwD in healthcare settings can contribute to the wellbeing of PwD during their stay in a hospital, according to the available literature published [25]. Researchers understood that publications could have been generated within specific contexts either as a white paper or as a protocol or guideline format. Scoping reviews are particularly useful to answer a broader and more flexible question, characterized by elements of the population, concepts, and contexts [26]. The search was divided in two subphases: (1) dementia care pathway search and (2) dementia-friendly hospital search. Three different databases (Pubmed, Scopus and Web of Science-WoS) were used to analyze evidence published in English and Spanish during the second quarter of 2020. Two researchers independently reviewed titles and abstracts of the identified articles. Researchers did not use any software for collecting data from manuscripts. Inclusion criteria were studies related to dementia care pathways and dementia-friendly hospital environments that were published in peer-reviewed journals. The inclusion criteria for the grey literature search were national and international associations, societies, and entities that support PwD and families.

Results gathered from this review may support healthcare staff to identify the critical points of healthcare institutions to be transformed into dementia-friendly settings. It is also part of a broader project that aims to design and pilot a dementia care pathway into hospital settings in Spain toward improving the wellbeing of PwD and their family caregivers during hospitalization.

### 2.1. Data Extraction and Analysis

A standardized form was used to summarize the content of each article. The variables extracted were authors, year, topic, place, participants/studies, type of study, methods and findings. Search results for all databases were compiled. Duplicates and nonrelated papers were excluded. Titles and abstracts of the remaining papers were assessed against the inclusion and exclusion criteria independently by both authors. The resulting papers were clustered, and disagreements were resolved through discussion based on the full text article. 

### 2.2. Study Selection

#### 2.2.1. Subphase 1. Dementia Care Pathway Review

The first step consisted of a literature review about clinical pathways in relation to PwD by using the following keywords in English and Spanish: (“dementia” OR “Alzheimer”) AND (“care pathway” OR “integrated care pathway” OR “critical pathway” OR “clinical pathway”). The exclusion criteria were studies not directly or indirectly related with dementia as well as processes and procedures focused on non-dementia syndromes, interventions related to pharmacological treatments and those not oriented to hospital pathways. The inclusion criteria were studies directly or indirectly related with the evaluation, implementation and design of care pathways for PwD in healthcare settings. There was no limitation in years and other categories. A total amount of 19 research articles were selected and reviewed in depth (Figure 1).

#### 2.2.2. Subphase 2. Review of Evidence about Dementia Friendly Hospital 

At this part, the keywords used in English and Spanish were: (“dementia-friendly”) AND (“dementia” OR “Alzheimer”) AND (“hospital” OR “clinical setting” OR “healthcare setting”). The searches were completed in the same databases: PubMed, Scopus and WoS. Inclusion criteria were studies that discussed the implementation and evaluation of dementia-friendly actions directly or indirectly related with healthcare settings. The exclusion criteria were those articles not directly or indirectly related with healthcare settings. There was no limitation in years and other categories. A total amount of 17 research articles were selected and reviewed in depth (Figure 2).

#### 2.2.3. Grey Literature Review

Guidelines and Protocols about dementia care interventions in different countries as well as white papers, best practice statements and reports from scientific organizations and funded projects regarding the topic were searched. Web based resources, reports and other documents were explored in order to complete the findings from the databases. The keywords were introduced in Google and Google scholar. The search was conducted in Spanish and English, collecting complementary resources to be used in the later design of the care pathway.

## 3. Results

### 3.1. Dementia Care Pathway

Most of the published literature about clinical pathways are focused on the medical approach of diagnosis and treatment [27,28,29] while others are focused on the statistical impact of dementia in health economics [13,30]. It is due to the differences between countries and institutions about precise diagnosis, treatment and care based on the healthcare system organization in the different levels of care. Although it is a relevant issue, it is not the main scope of this review. 

One of the most important issues about PwD admission to clinical settings is the length of stay in hospitalization, as it is directly related with unintentional harmful events. However, little attention is paid about what determines wellbeing and length of stay in hospitals for PwD in care pathways. According to the findings, the key points that directly impact hospital stay of PwD are: medication for neurological, cognitive and psychiatric disturbance, visits to other departments within hospital, and invasive procedures [31,32]. But facilitators that contributed to the early discharge of the patient were: family members living with PwD, caregivers supporting the primary caregiver and living in their own home. Another identified element in literature that can contribute to the earlier discharge from hospital facilities of PwD was staff organization. Successful activities organized by those interdisciplinary teams were: establishing a diagnosis, regular meetings led by physicians & nurses with the caregiver & patient to explain future and daily tasks, cares and other activities during hospital stay [33,34,35]. In those examples, nurses are involved in education and symptom management or behavior problems more than physicians, which results in an important role when talking about building confident relationships [36]. 

One of the most common situations lived by PwD in hospital settings corresponds with the last stages of their lives. Authors stressed end-of-life process as a relevant topic in literature when searching information about dementia care pathways. Scientific societies that support this issue advocate for (a) clear communication, (b) compassionate care, (c) multidisciplinary approach and (d) debriefing sessions for staff. Clear communication is needed in the last period of life from staff to relatives, tutors and the affected person. It facilitates decision-making in accordance with the person’s wishes and needs, including the possibility of choosing the settings, considering the situation of the person. Compassionate care is the gold standard when performing palliative care and healthcare staff should be trained in order to be able to understand PwD and relatives’ needs. This care should be performed by multidisciplinary teams, whose members need to be trained in approaching the person and relatives from different perspectives to ensure quality end-of-life care. This stage of care may increase stress in healthcare staff. Debriefing sessions may help them confront and discuss the lived situation to perform new strategies as a process of continuous improvement [37,38,39,40]. As it has been mentioned before, it is necessary to evaluate PwD from a multidisciplinary perspective and take decisions based on PwD wishes and last evidence. Thus, in advance stages of the disease, medical interventions may not improve quality of life of PwD and neither extend years of life but may cause more suffering to PwD and caregivers along the process such as using PEG feeding [31]. It is a relevant issue as not all countries and rural areas count with structured and equipped healthcare services to attend end-of-life processes at home, and therefore they live their last days in hospital settings.

But barriers should be considered when implementing new initiatives for PwD in healthcare facilities. The negative indirect outcomes found by implementing dementia CPs was that staff usually reports higher workload as processes take longer, delaying other activities. Another common issue found is the lack of primary care coordination. Prevention is not performed equally in all cultures and contexts, thus adapting processes to prevent unnecessary hospital admissions requiring collaboration from primary care institutions forcing unnecessary admissions of PwD in hospital settings. At the same time, many difficulties have been described about care coordination between outpatient and inpatient services, as aligning goals between different healthcare facilities is a challenge in healthcare systems worldwide [41,42].

Evidence published specifically about Dementia CPs in Spanish were not found. Table 2 Dementia Care Pathway Search shows the literature findings about the dementia care pathway.

### 3.2. Dementia-Friendly Hospital

As stated by Innes [43] in a comparative prospective study including PwD and healthcare staff, dementia-friendly interventions in hospital settings should start from changing the culture of the institution. Staff training and architectonical adaptation to PwD are specific actions to take when changing the culture of institutions to be more dementia inclusive. Following the perspective of Waller & Masterson [44], the key elements to be considered when adapting an acute hospital environment should encourage meaningful activities, including caregivers, fostering easy decision-making based on PwDs’ wishes. Regarding signposting, legibility (big letters with colorful contrasts), wayfinding, and familiarity of the environment contribute to PwD independency, safety and reduce agitation during hospital stay. Dementia-friendly program implementation in healthcare settings have the potential of preventing aggression and incidents in healthcare settings. In order to adapt the physical environment, authors outlined the following items in Figure 3 Key elements for adapting PwD environments [45,46,47]. 

Not only indoor changes are required when adapting hospital settings to PwD. Outdoor hospital adaptation has to be considered when adapting healthcare settings. Public transport, easy entrance, and physical barriers that surround the hospital infrastructure influence the environment to be more inclusive with PwD and caregivers. Car parking and lack of outdoor seating were difficulties that participants remarked as important for them when accessing hospital facilities in different literature reviews and case studies [48,49], although outdoor spaces, gardens or green and natural spaces have also shown therapeutic effects on PwD, as they promote active engagement and physical and social interaction [50,51,52]. 

Another important issue when adapting practice to become more friendly with PwD is that staff has to be trained in order to recognize, care and treat signs and symptoms as stated by Palmer [53] in a prospective intervention focused on an educational intervention with 355 healthcare staff. A considerable amount of PwD develops an episode of delirium that can be manifested by “alertness, poor concentration, disorientation, behavior and speech changes, changes in sleeping patterns, mood swings, paranoia and hallucinations” [54], while hospitalized, usually deriving in late hospital discharge and in satisfaction. The impact of a hospital or other healthcare settings not being adapted to PwD is not only negative when the patient accesses the building, but also once the person is hospitalized. In the Okamura study [55], 78.8% of a sample of 74,171 participants over 65 years old responded that they felt anxious about having dementia and not being properly cared. This still occurs when people are diagnosed with dementia, as evidenced by Møller [56] in a qualitative study in which PwD repeated the sentence “if only they could understand me”. Those people believed that staff were not employing enough time to know them and their actions were misunderstood under the umbrella of being diagnosed with dementia. Further than knowing how to care for the different diseases that involve dementia, authors found that healthcare staff needs to develop and train more than just skills and competencies to care for PwD properly but also attitudes for acting one step beyond and elaborating individualized care plans, which are related to compassionate care principles [57]. These added values are related to dementia empathy and an understanding behavior as a positive way of communicating unmet needs, staff experiential learning promotion, support from clinical experts in dementia care, individualized care, psychosocial interventions and confidence for risk management. All these identified values are necessary to be included as core elements when training healthcare staff.

The latest qualitative publications about how to transform a hospital into a dementia-friendly environment outline the importance of the presence of caregivers in daily hospital routines while caring for PwD. Caregivers can act as expert advocates for PwD, contributing with staff to find solutions for best caring [54]. At the same time, other research studies evidenced the importance of practicing meaningful activities toward promoting self-realization and keeping the brain active. These tasks are not only related to self-care but also to cognitive stimulation. Chess practice has been investigated as one of the most stimulating tools for PwD, although it may be difficult for people that are in moderate to advance stages [58].

To achieve all these mentioned adaptations in healthcare settings is necessary to involve different players, not only those who are directly involved in caring for PwD such as healthcare staff and caregivers. Thus, other reviews about dementia-friendly interventions in hospitals have highlighted the importance of the role of the changing agents. The main characteristics identified for this role are (1) supportive peer facilitator, (2) organizational authority, and (3) clinical expertise. These roles need to be present when changing a complete process in a hospital in order to have a positive effect not only in the clinical approach but in the architectonical, economic and cultural situation of the institution once it has been decided to become a dementia-friendly environment [59,60]. Table 3: Dementia Friendly Hospital Search shows the search findings.

### 3.3. Dementia Care Pathway & Dementia Friendly Hospital Grey Literature Search

While searching for evidence about how to implement a dementia care pathway in a hospital setting, grey literature emerged as useful information, although none came from the Spanish context. Published care pathways for dementia are usually focused on specific areas such as end-of-life care or primary care [61,62], although less about hospital pathways. However, clinical guidelines and websites detail information about the steps that the PwD and caregiver should follow before, during and after visiting a hospital. Guidelines about how to prepare for a surgery in the case of PwD, what to bring in case you are admitted to the hospital, or how to facilitate the work of hospital staff are available on certain platforms that may also be helpful for other healthcare institutions when building a dementia-friendly culture [63,64]. Institutions such as Alzheimer’s Scotland [65] has available tools that help PwD and caregivers when they are admitted to healthcare facilities. Knowing the person that staff is caring for is important in order to promote person-centered care instead of impersonalized care. Alzheimer Scotland introduced the tool “Getting to know me” in order to collect relevant information about what really matters to the cared person, in order to have different tools and resources to perform individualized care, based on information provided by the patient or the caregiver.

National and international guidelines found were mostly based on the implementation and practical information when designing a dementia care pathway. Relevant guidelines were found such as the ones by Grey [66] and Halsall & Mcdonald [67] designed for healthcare centers and other living spaces for PwD and paying special attention to users’ feedback and architectonical elements. However other institutions have published information focused not only on the design but in the care process such as by the National Collaborating Centre for Mental Health [68]. It remarks dementia key principals from NHS England’s Well Pathway for Dementia (diagnosing, living, supporting and dying well). Under this paradigm, a complete dementia care pathway was developed following the main quality statements of NICE Quality Standards on dementia [69]. Seven quality statements are the mandates for caring for PwD: (1) raising awareness-health promotion interventions, (2) diagnosis, (3) advanced care planning, (4) coordinating care, (5) activities to promote wellbeing, (6) managing distress, and (7) supporting caregivers. Additionally, the Palliare Best Practice Statement used by the Alzheimer’s Europe and produced by the partners of a European funded Project called Palliare Project as a white paper was included in our search. This document reflects about six overarching principles used as a reference when caring for PwD [70]. It advocates for the rights, advanced dementia care, symptoms management, quality of life, family support and advancing practice. 

## 4. Discussion

According to the literature reviewed, different, although not many, initiatives have been conducted in practice in relation to the development of dementia care pathways worldwide. PwD are differently cared for depending on the country, context and culture. Thus, available resources may be different in order to perform the best care possible that directly depends on the level of professional health care training, settings, available resources, ratios patient per healthcare staff, awareness about dementia, and government support. Staffing levels through the UNISON survey [71] found that 66% of nurses did not have sufficient time to care properly for patients. They thought that ratios of eight or more patients to one registered nurse (RN) resulted in poorer patient outcomes and higher nurse stress. It has been highlighted as one of the most difficult issues to address when approaching care excellence in dementia care, such as Spanish nurse ratios in hospitals are 12.7 patients per nurse [72]. Due to this high workload for healthcare staff in hospitals, caregivers and families play an important role in PwD care within clinical contexts by contributing to the daily basic care such as hygiene, feeding, oral treatment and supervision to avoid physical restraints. As important as to know how the disease affects the brain, it is important to facilitate the decision-making during hospital stay in order to avoid harmful events to the maximum. Caregivers still play an important role in several Hispanic–Latin cultures and therefore need to be taken into consideration when adapting a hospital environment, as they can sometimes express and act on behalf of those affected by dementia in moderate to severe phases of diseases. Caregivers do not only help staff in understanding PwD, but they also contribute to stimulate the patient from a nonpharmacological intervention, using resources such as mindfulness, physical and cognitive stimulation [73]. This means that this theoretical approach of adapting institutions to PwD still seems to be ideal in certain contexts as evidence is also showing that insufficient staff in hospitals will bring negative effects to PwD and other patients hospitalized as higher workload will be present [74]. 

One of the critical points when adapting healthcare settings to become more dementia-friendly are the standards and evaluation methodology. Along the review, little has been mentioned about how to measure if a healthcare setting meets a minimum criteria to ensure it is a dementia-friendly environment. But implementing best practices from different studies into specific context is not enough. However, the most used tool to transform healthcare settings into dementia-friendly is the King’s Fund tool [44]. Although it is mainly focused on the architectonical design, it provides a clear guideline for designing and evaluating a healthcare setting. It has been widely used in the United Kingdom but initiatives in different European contexts are being piloted [75]. 

The vast majority of dementia care pathway interventions included in this study have followed similar structures in which communication, coordination of care, evaluation and use of appropriate resources stimulate the transformation of a healthcare setting into a dementia-friendly environment. One of the critical points when adapting and changing the culture of an institution is to identify the changing agents [59]. These main actors have the capability to influence at the different hierarchical levels of the institution in order to facilitate the implementation and use of resources needed to adapt the hospital environment to PwD. As an example of easy decision making in healthcare institutions, those identified as “flat organizations” are used as an example of management in which the implementation of dementia care pathways and dementia friendly initiatives can be easily implemented. Parallel to this situation, those institutions in which evidence-based practice is not fostered and continuously updated, there is a strong hierarchical management structure and culturally there is no support to build dementia inclusive contexts. Thus, it becomes more difficult to successfully implement these initiatives. 

The paradigm named as prudent healthcare has been found in literature when talking about efficiency and fostering dementia inclusivity settings [76]. It is clear that improving dementia care is still not a priority in certain cultures due to the lack of specific implementation of dementia plans worldwide. The COVID-19 health, social, economic and political crises make this issue more complex. However, it is important to remark that PwD are one of the most vulnerable population. During hospitalization, they should not be treated only as patients infected by COVID-19. Risk of delirium, self-harm and behavior-difficult situations may cause worsening in dementia disease and COVID-19 infection during isolation. Thus, a different approach and care pathway should be developed in healthcare institutions in order to build a response to COVID-19 in PwD [77]. Several authors have shown their findings highlighting the importance of investing in dementia workforce training, changes in thinking and leadership, continuous improvement, respecting and engaging the workforce as valuable contributors of new ideas, and vigorously pursuing the needs and experiences of service users [78,79]. At the same time, it has been clearly stated the importance of architectonical adaptation in terms of light, floor and furniture, in order to avoid daily routines disturbances [80]. But the combination of the core values promoted by prudent healthcare and dementia friendly paradigms can potentially be useful for all healthcare institutions as it aligns with the strengths of focusing on dementia care but optimizing the available resources with special attention to staff training between other interventions mentioned before. This approach can be an efficient strategy to improve dementia care within available resources in order to perform little steps toward achieving dementia awareness in the different healthcare systems worldwide.

## 5. Limitations 

Despite the intention of including all relevant literature, we assume the possibility that studies and relevant grey literature may not have been identified as it is a limitation in scoping reviews. Furthermore, this review was not intended to be an exhaustive review of the literature but to concentrate on the key themes and necessary elements to focus on when starting to change dementia-friendly healthcare environments. This review includes studies from different methodological designs (qualitative and quantitative studies, questionnaires, reviews, reports, case studies, etc.). Another important limitation is the non-homogeneous time of observation from different geographic areas in the different included studies of this review. Although we have tried to include mostly papers based in hospital settings, studies that have not been conducted in hospitals were included, as it clearly shows interventions that can be potentially applied to any healthcare facility. Thus, this paper brings an overview perspective of the first steps to consider when designing a dementia inclusive intervention in healthcare settings. 

## 6. Conclusions

Developing CPs is a complex, iterative process. Integrated care pathways should be practical, adaptable to different care contexts, and lead to improved patient, family, and staff outcomes. This type of approach will help staff personalize interventions for individuals with dementia and it will make future institutions developing a framework to get to know the patients better, from an individual perspective [81]. Thus, care quality will increase as the current paradigm is changing to become more person-centered. Person recognition, collaboration, and validation of their reality and supporting PwD to live a life with dignity will contribute to switch the current meaning of being hospitalized. This paper summarizes the bullet points to focus on when adapting a healthcare setting to PwD, which are changing the mindset of the staff by developing a clear path that can guide PwD admissions into hospitals, and moreover, the importance of the environment in facilitating or worsening the situation of PwD and their caregivers. According to the NICE [69] quality guidelines, CPs should be focused on end-of-life processes, dementia care training for healthcare, and nonhealthcare staff in order to develop dementia awareness and support for caregivers during admissions and stimulating PwD. Architectonical adaptation needs to be considered, specifically in terms of light, floor, furniture, noise, wayfinding, signposting when referring to indoor settings and green and natural areas, easy entrance, architectonical barriers, and comfort referring to outdoor areas. 

Dementia care pathways are involving inclusiveness for PwD and families in developing new ways of care in which joint decisions can be found in order to build a collaborative working strategy. Future research should concentrate on evaluating the implementation of dementia-friendly care pathways in order to find differences and similarities among healthcare settings to help building dementia-friendly healthcare awareness. This literature review may be a readily scalable example for enabling hospitals and healthcare staff to foster dementia care excellence based in the PwD recognition, which can result in significant physical and psycho-social benefits for those living with Alzheimer’s and other dementia diseases and their caregivers.

## Figures and Tables

**Figure 1 ijerph-18-09296-f001:**
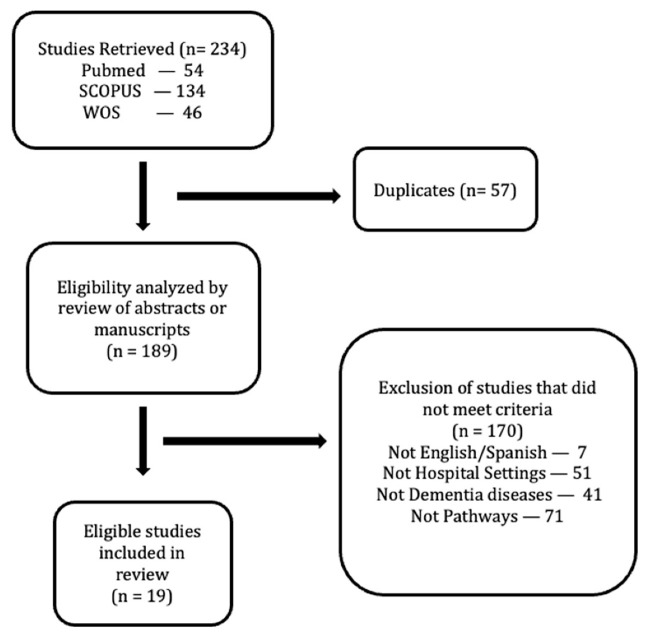
Inclusion search flow diagram for studies in dementia care pathway in this review.

**Figure 2 ijerph-18-09296-f002:**
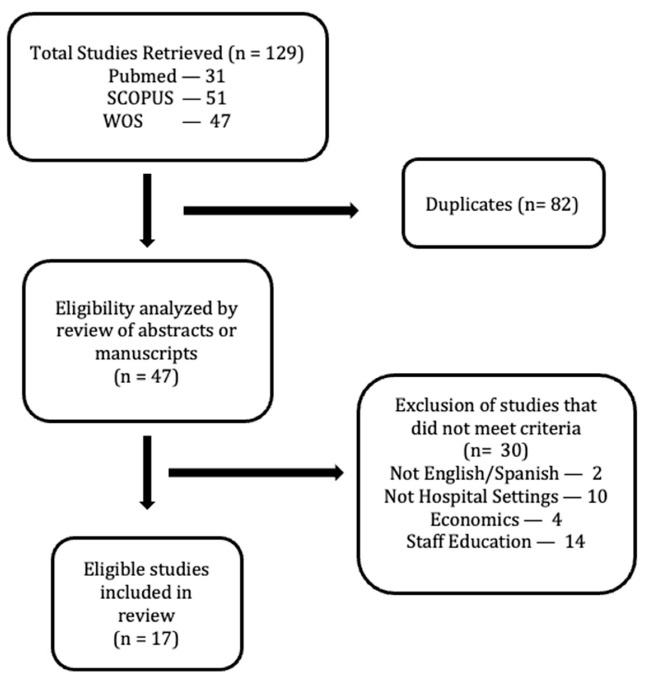
Inclusion search flow diagram for studies in dementia-friendly hospital adaptation in this review.

**Figure 3 ijerph-18-09296-f003:**
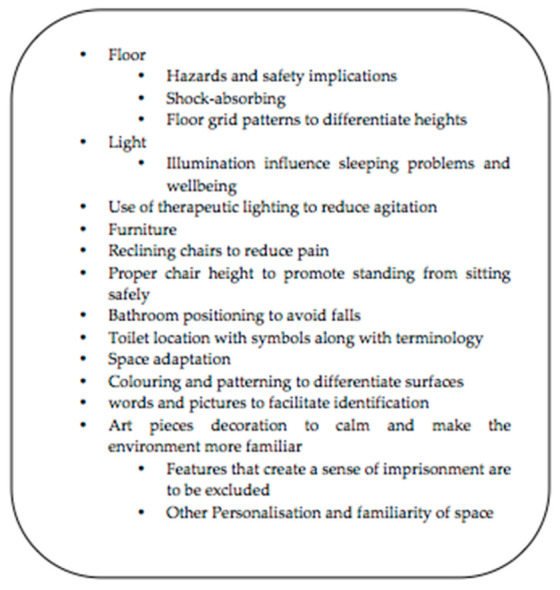
Key elements for adapting PwD environments.

**Table 1 ijerph-18-09296-t001:** Countries with a national dementia plan [18].

Australia	France	Malta
Austria	Greece	Mexico
Canada	Indonesia	Netherlands
Chile	Ireland	Norway
Costa Rica	Israel	Qatar
Cuba	Italy	Slovenia
Czech Republic	Japan	Switzerland
Denmark	Republic of Korea	UK
Finland	Luxembourg	USA

**Table 2 ijerph-18-09296-t002:** Dementia Care Pathway Search.

Authors/RN	Year	Topic	Origin	Participants-Studies	Type of Study	Methods	Findings
[27] Filippini et al.	2004	Implementation of an integrated care pathway for dementia in clinical practice	Italy	Patients with suspected dementia syndrome	Prospective cohort study	Implementation of the ICP including education of GPs	The ICP implementation improved the appropriateness of patients’ referral to specialists.
[28] Kua et al.	2007	A two-year review of the utility of dementia clinical pathway in a psychogeriatric inpatient	Singapore	PwD admitted into the acute psychogeriatric wards in 2005–2006, who fulfilled ICD-9 criteria	Retrospective analysis review	Database hospital extraction and analysis	CP can serve as a template to co-ordinate and document inpatient dementia care by the multidisciplinary team.
[29] Brodaty & Cumming	2010	Dementia services	Australia	PwD in Australia	Limited review of current government policies and relevant papers.	Data extraction from relevant documents	Country services need to be more accessible, flexible and provided in a timely manner to respond effectively to the diverse needs of PwD and their caregivers.
[13] Kazui et al.	2004	Effectiveness of a clinical pathway for the diagnosis and treatment of dementia and for the education of families	Japan	43 PwD	Intervention and control group	Primary caregivers, physicians, and nurses were given a questionnaire to obtain their comments about the impression of treatment with the CP	The CP deepened the caregiver’s understanding of the sequence of medical practices for the inpatient, the disorders of the inpatient, the treatment methods, and the methods for coping with the disorder.
[30] Gervasi et al.	2019	Integrated care pathways on dementia	Italy	Two raters	Survey Analysis	Checklist based on the GICPD	Policy- and decision-makers should pay more attention to the GICPD when producing ICPs.
[31] Bouza et al.	2019	Effect of dementia on the incidence, short-term outcomes, and resource utilization of invasive mechanical ventilation in the elderly	Spain	259,623 cases identified	Retrospective population-based study	Descriptive and comparative analysis of cases	PwD had a 5.39% higher incidence of mechanical ventilation in average annual than people without dementia
[32] Traynor, Britten and Burns	2016	Developing delirium care pathway	Australia	Registered practitioners (*n* = 45)	Qualitative approach	Focus groups and one-to-one interviews	Importance of the layout, color and clarity of the new pathways.
[33] Coleman	2012	End-of-life issues in caring for patients with dementia	United Kingdom	-	Review of guidelines and evidence on palliative management	Commentary	Active care to palliative care, considering the longitudinal progression of dementia illness is a challenge is dementia care.
[34] Ghiotti	2009	End of Life Care Project Supporting families caring for people with late stage dementia at home	United Kingdom	40 families	Prospective Intervention study	Questionnaires	The involvement of GP’s, district nurses and other community staff is essential to support people with late stage dementia at home and to avoid unnecessary admissions to hospital.
[35] Volpe et al.	2020	Pathways to care for people with dementia	15 countries	548 consecutive clinical attendees with a standardized diagnosis of dementia	International multicenter study	Semi structured interviews	Variation in clinical dementia management worldwide in terms of treatment, delay to specialist dementia care and the first problem presentation is memory issues followed by psychiatric symptoms.
[36] Sampson et al.	2005	Efficacy of a palliative care approach in advanced dementia	United Kingdom	-	Review	Systematic review	The assessment of outcomes for end-of-life care in patients with dementia is methodologically difficult.
[37] Waterman, Denton, Minton	2016	End-of-life care in a psychiatric hospital	United Kingdom	Person with Dementia	Case study	Single Case	Importance of early advance care planning as soon as dementia is diagnosed. A barrier to early advanced care is that families and healthcare staff may not always recognized dementia as a terminal illness.
[38] Martin, O’Connor & Jackson	2020	Gaps and priorities in dementia care in Europe	Europe	-	Literature Review	Scoping Review	Narrowing the gaps to improve care experiences and the support for people living with dementia care encompass person-centered care, integrated care pathways, and healthcare workforce development.
[39] Minghuella & Schneider	2012	Rethinking a framework for dementia care	United Kingdom	-	Qualitative	Conceptual discussion	People living with dementia have assets as well as needs; people as partners in care, nurture their capacity and capabilities.
[40] Afzal et al.	2010	End-of-life care for dementia patients during acute hospital admission	Ireland	75 patients with dementia	Qualitative	Retrospective clinical case note review	Dementia patients are significantly less likely to be referred to palliative care interventions, to be prescribed palliative drugs and to have caregivers involved in decision-making.
[41] Tang et al.	2017	Gaps in care for patients with memory deficits after stroke and risk of future dementia	United Kingdom	17 Healthcare providers	Qualitative study	Semi-structured individual interviews	Less focus on memory and cognition in post-stroke care, difficulties bringing up cognition and memory problems post-stroke, lack of clarity in current services and assumptions made by healthcare professionals introducing gaps in care.
[42] Tropea et al.	2017	Caring for people with dementia in hospital	Australia	112 Multidisciplinary healthcare staff	Qualitative study	17-item survey	The environment, inadequate staffing levels and workload, time, and staff knowledge and skills were identified as barriers to implementing best practice dementia care.

**Table 3 ijerph-18-09296-t003:** Dementia Friendly Hospital Search.

Authors	Year	Topic	Origin	Participants-Studies	Type of Study	Methods	Findings
[43] Innes et al.	2016	Living with dementia in hospital wards	Malta	Sixteen patients with dementia and 69 staff	Comparative prospective study	Environmental audit tool, Dementia care mapping, Staff questionnaires, Validity and reliability	Signs, natural lighting, access to outside spaces, respecting their rights and dignity, avoiding restraint improved care experience.
[44] Waller & Masterson	2015	Designing dementia-friendly hospital environments	United Kingdom	26 NHS trusts in England	Qualitative	Case study	Simple, cost-effective changes to the physical environment of care have positive effects on PwD reducing agitation and distress and raising staff morale.
[45] Parke et al.	2017	Scoping Literature Review of Dementia-Friendly Hospital Design	NA	28 primary studies plus expert reviewers’ narratives	Review	Scoping literature review	Physical design influences the usability and activity undertaken in a health care space and ultimately affects patient outcomes.
[46] Waller, Masterson, Evans	2016	Assessment tools to support the creation of dementia friendly care environments	United Kingdom	26 NHS trusts in England	Qualitative	Assessment tool development and evaluation	Meaningful interaction between patients, their families and staff, well-being, eating and drinking, mobility, continence and personal hygiene, orientation, calm, safety and security contribute to improve hospital wards for PwD.
[47] Parke & Hunter	2017	Dementia-friendly emergency department	Canada	NA	Qualitative	Report	Clinical care systems and processes, social climate, policies and procedures, physical design & communication tools to be considered for designing dementia friendly wards.
[48] Van der Berg	2020	Barriers and Enablers to Using Outdoor healthcare Spaces	NA	Twenty-four studies were included	Review	Systematic Review	Design of outdoor areas, pathways and surfaces, seating, shade and shelter, proximity and social activities were identified as enablers.
[49] Xidous	2020	Key Issues for Patients and Accompanying Persons	Ireland	95 questionnaires to patients and/or APs, and conducted 12 structured interviews	Qualitative prospective	Case Study	Lack of direct access from public transport, traffic volume and parking, main entry doors, wayfinding and orientation challenges within the hospital, wheelchair availability and noise levels were some of the highlighted issues.
[50] de Boer et al.	2017	Green Care Farms as Innovative Nursing Homes	Netherlands	115 nursing home residents at baseline	Qualitative prospective	Ecological momentary assessment	People living in green places significantly participated more in domestic activities and were physically more active and practice outdoor/nature-related activities than nursing home residents.
[51] Bossen	2010	Outdoor Nature and dementia	NA	NA	Review	Unstructured review	Nature interaction with PwD provides a natural multisensory stimulation, improving quality of life and dignity of PwD.
[52] Brawley	2007	Designing Outdoor Spaces for Individuals with Alzheimer’s Disease	NA	NA	Review	Unstructured review	Outdoor scenarios need to promote Exercise and Physical Activity through natural light and gardens adapted to Pwd to promote safety and security mobility, as well as fall prevention.
[53] Palmer et al.	2014	Dementia Friendly Hospital Initiative education program for healthcare staff	USA	355 healthcare staff members	Prospective	Questionnaires	Shared decision making and understanding the needs of these patients and their caregivers improve satisfaction
[54] Fitzpatrick	2018	Strategies to develop dementia-friendly hospital wards	United Kingdom	-	Qualitative	Commentary	High nursing ratios negatively affect dementia quality care in hospitals. Nursing training to identify early symptoms of delirium. Caregivers’ involvement in care. Background information collection to know better the PwD.
[55] Okamura	2019	Dementia care needs	Tokyo	74,171 participants	Prospective Quantitative	Questionnaire	Not having someone who can take you to the hospital when you do not feel well, not trusting in neighbors, higher educational level (>9 years), not having someone to consult when you are in trouble, not working and other issues were identified as producing anticipatory anxiety.
[56] Møller et al.	2018	Acute hospital care experiences of patients with Alzheimer’s disease	Denmark	3 people with Alzheimer’s disease	Prospective qualitative	Participant observation	Involving patients into their daily care activities, self-blame of their current situation and appreciation of the staff are some of feelings experienced by people affected by Alzheimer’s disease in hospitals
[57] Iaboni et al.	2020	Compassionate care in isolated PwD in healthcare centers	NA	A person with dementia	Qualitative	Commentary	A clear structured strategy is needed to stimulate isolated Pwd using music, coloring, worksheets and individualized care plan.
[58] Lillo-Crespo et al.	2019	Protective Factors in Dementia	NA	21 studies	Review	Scoping review	High Mental Activities such as Chess may contribute to prevent Dementia worsening
[59] Handley, Bunn & Goodman	2017	Dementia-friendly interventions to improve the care of people living with dementia admitted to hospitals	United Kingdom	Phase 1 combined findings from 15 stakeholder interviews and 22 publications to develop candidate program theories. Phases 2 and 3 identified and synthesized evidence from 28 publications	Review	Realist review	Understanding behavior as communication, experiential learning and creating empathy, adapt working practices and routines to individualized care, psychosocial needs, building staff confidence to provide person-centered risk management.
[60] Brooke & Semlyen	2019	Impact of Dementia-Friendly ward environments	United Kingdom	Participants were junior qualified nurses (17) and health care assistants (21)	Prospective Qualitative	Focus groups	Individualized care plan, physicial environment adaptation to PwD need and staff resistance to change.

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
