# Peer review of "Designing Dementia Care Pathways to Transform Non Dementia-Friendly Hospitals: Scoping Review"

_ijerph, 2021, doi:10.3390/ijerph18179296_

Round 1

Reviewer 1 Report

I have been review before this paper in this journal. The autors applied all the suggestions  in the paper. The work is now much better. I think is ready to continue with the  editorial process.

The article support dementia area and can improve knowledge in health projects.

Author Response

please find the attachment, thanks for your valuable comments.

Reviewer 2 Report

The review carried out follows the guidelines established by PRISMA and the Joanna Briggs Institute methodology. Its results are valuable for the generation of new care strategies, not only in hospital settings, but also in broader health settings and towards the construction of “a health awareness favorable to dementia”.

Author Response

(The authors gave the same response as above.)

Reviewer 3 Report

Designing Dementia Care Pathways to Transform non Dementia-Friendly Hospitals: Scoping Review

Thank you for submitting your work to the International Journal of Environmental Research and Public Health, and providing me with the opportunity to review your paper. I hope you find my comments both helpful and constructive as this is my aim.

Unfortunately there are grammatical and sentence construction errors throughout your work that impact on meaning, these need to be addressed. I have highlighted some errors below, but not all.

It is important to recognise that the word dementia, when is a sentence does not have a capital D.

Abstract

  • In the first sentence the word ‘nowadays’ is not required, please remove
  • Unfortunately the second sentence contains poor grammar and is difficult to follow, please do not refer to people with dementia as ‘they’, either in your abstract or throughout your paper
  • Unfortunately there are grammatical errors which impact on the meaning of sentences within the abstract as a whole and these need to be addressed for clarity, and it is unclear how you recommendations follow from your findings
  • The last line of your abstract is an overarching conclusion, which needs to be amended to represent your findings and discussion, please do not over state these

Background

  • In the first sentence there are grammatical errors and the world ‘alarming’ is not appropriate in academic writing
  • Neurodegenerative diseases are very broad, where is the data explicitly referring to dementia
  • PwD needs to be in full the first time applied in your manuscript

Dementia care pathways

  • Why are you discussing ‘brain impairment’? This is a paper on dementia, so it is important to apply the same language and terms throughout
  • This is not an ‘uncomfortable situation’, this is inappropriate language to use when discussing dementia, as uncomfortable for whom? This term needs to be removed
  • Doble – spelling error? Double
  • Is Table 1 necessary?

Dementia friendly interventions

  • What is meant by ‘rewell known regions’?
  • Also in the above sentence what do you mean by ‘norms’? Further clarity is required
  • This section is rather disjointed and doesn’t follow smoothly to the aim of your review, further amendment is required

Materials and Methods

  • In the first paragraph some of the basic information explaining the purpose of a scoping review can be removed and more information on your two searches, the dates of publications included needs to be described not the date the search was completed, this is important

Data extraction and analysis

  • There is no information in this section regarding analysis, how was it decided to present the information in the current format?

Dementia Care Pathway

  • An overview of the papers included in this section is required to provide the reader with some understanding of the papers you are discussing, such as studies, countries, white papers, this information should not just be presented as a table at the end of this section, this would support and direct the information provided in this section
  • Unfortunately this section is difficult to follow due to grammatical and sentence errors, and no clear information has been provided regarding dementia care pathways
  • Why did the authors stress the importance of end-of-life care? Which authors are you referring to? End-of-life care for people with dementia is unusual in acute hospital care settings in many countries, further clarity is required here

Dementia Friendly Hospital

  • There is the need for an overview of the papers included in this section to provide the reader with some understanding of the papers you are discussing
  • You have not mentioned the King’s Fund, who have completed a large amount of work on this topic in the UK
  • You reference one study and results but this does not include people with dementia, why is this here?

Why have you presented the data from research papers and grey literature separately, if this is the case did you identify he robustness of the research papers you included?

Discussion

There is a need for clearer presentation of the results section to inform the discussion and then the development of the conclusion

Author Response

(The authors gave the same response as above.)

Reviewer 4 Report

This study addresses an important and timely issue. However, the manuscript needs a lot of improvements to show the results and provide a clear message. I think that this work has a lot of potential, but it is not in its final form to be submitted and published in this journal.

Author Response

please find the attachment, thanks for your valuable comments.

This manuscript is a resubmission of an earlier submission. The following is a list of the peer review reports and author responses from that submission.

Round 1

Reviewer 1 Report

This article is  very interesting, and contributes to the development of scientific knowledge.I have some suggestions.

1.     Introduction
This  scoping review  is part of a project that aims to design and pilot a dementia care  pathway into hospital settings in Spain to improve the wellbeing of PwD and their family  caregivers during hospitalisation ., but also of results  of this article can contibute to the scientific community. Dementia is a global problem, I believe that autors can be expanded the scope and not make it seen only as part of a research Project. Could make a better justification  of this point

2.    Methodology 
a. Is not clear how autors conduct and report scoping review, there are some method like (JBI)
b.    The time interval where they searched for the articles is not clear. The last 10 years?
c.    Which combine keywords with connectors AND and OR  the autors used?
d.    Which resources autors used to evaluated the qualite of research of each one of the  articles?e.    Grey literature can be difficult not only to find but also to evaluate. It usually has not had the same quality checking as peer-reviewed material, so careful evaluation of the source. How the authors make this, is n´t visible in the article. In fact on the article table  han been incluyed some of this grey paper found.

e. Data should be collected in the figure chronologically  for each included study

  Conclusion 
My suggestios is summarize research findings, to identify research gaps, and to make recommendations for the future research.

Author Response

Dear Reviewer, 

I highly appreciate your comments and suggestions. I believe it really contributed to improve the manuscript we are building. Below you will find the answer to your comments: 

  1.    Introduction
    This  scoping review  is part of a project that aims to design and pilot a dementia care  pathway into hospital settings in Spain to improve the wellbeing of PwD and their family  caregivers during hospitalisation ., but also of results  of this article can contibute to the scientific community. Dementia is a global problem, I believe that autors can be expanded the scope and not make it seen only as part of a research Project. Could make a better justification  of this point. Included in Introduction- 1.3 Dementia-friendly interventions - Last paragraph
  2.    Methodology 
    a. Is not clear how autors conduct and report scoping review, there are some method like (JBI). PRISMA guidelines were followed
    b.    The time interval where they searched for the articles is not clear. The last 10 years? No limitation with years was stablished as there were some articles found as relevant more than 15 years.
    c.    Which combine keywords with connectors AND and OR  the autors used? It has been added in setion 2.2 Study selection
    d.    Which resources autors used to evaluated the qualite of research of each one of the  articles?e.    Grey literature can be difficult not only to find but also to evaluate. It usually has not had the same quality checking as peer-reviewed material, so careful evaluation of the source. How the authors make this, is n´t visible in the article. In fact on the article table  han been incluyed some of this grey paper found. PRISMA guidelines were followed
  3. Data should be collected in the figure chronologically for each included study. We followed the example of other scoping reviews published by the journal (order of mentioned studies in the manuscript)

      Conclusion 
    My suggestios is summarize research findings, to identify research gaps, and to make recommendations for the future research. It has been included in conclusions. 

All the best, 

Reviewer 2 Report

This scoping review explored the evidence of the requirements and characteristics for designing a Dementia Care Pathway in hospital settings. The review found that person-centred care, compassionate care and end-of-life process should be integrated into the design of a dementia care pathway along with modifications for people living with dementia in indoor and outdoor spaces in hospital settings.  Overall, the findings were interesting showed a need for more research in this area. It has implications for policy and practice. 

Comment 1: There were many grammatical and spelling errors making this manuscript very difficult to read. The manuscript needs to be reviewed again, possibly by a native English speaker. Extensive editing of English language and style required. 

Comment 2: Please be sure to explain your abbreviations when you first use them. You do not do this for CP or care pathways.

Comment 3: Was there any independent reviewing or checking of the studies included? This was not stated in the methods. Also, was any reference or citation searching undertaken? 

Commented 4: How was data extracted? Was a data extraction table used? How was the extracted data then analysed and synthesised? This needs to be clarified in the Methods sections.

Comment 5: This paper would benefit from following the reporting guidelines set out in PRISMA Extension for Scoping Reviews (PRISMA-ScR). The Checklist and Explanation can be found here: https://www.equator-network.org/reporting-guidelines/prisma-scr/

Comment 6: The strengths and limitations of the review are not discussed in the Discussion as well as recommendations for future research based on your findings.

Author Response

Dear Reviewer,

I highly appreciate your comments and suggestions on this paper. I believe it has really improved the manuscript. Please find below the answer to your comments:

Comment 1: There were many grammatical and spelling errors making this manuscript very difficult to read. The manuscript needs to be reviewed again, possibly by a native English speaker. Extensive editing of English language and style required. Changes have been made along the text.

Comment 2: Please be sure to explain your abbreviations when you first use them. You do not do this for CP or care pathways. It has been included in first paragraph of section 1.2

Comment 3: Was there any independent reviewing or checking of the studies included? This was not stated in the methods. Also, was any reference or citation searching undertaken? It has been included in Materials and methods section end of the paragraph

Commented 4: How was data extracted? Was a data extraction table used? How was the extracted data then analysed and synthesised? This needs to be clarified in the Methods sections. We have added a new paragraph called data extraction in materials section

Comment 5: This paper would benefit from following the reporting guidelines set out in PRISMA Extension for Scoping Reviews (PRISMA-ScR). The Checklist and Explanation can be found here: https://www.equator-network.org/reporting-guidelines/prisma-scr/ It has been followed and included in the paper.

Comment 6: The strengths and limitations of the review are not discussed in the Discussion as well as recommendations for future research based on your findings. Limitations have been included in section 5 and future research has been included in conclusions.

All the best, 

Reviewer 3 Report

Thank you for the opportunity to review this manuscript reporting findings from a scoping review of dementia care pathways and dementia friendly hospitals. Findings will be used to inform the development of resources for hospital settings in areas not currently focused on considerations for dementia care. The review identified a range of interventions and summarised the evidence. My comments are as follows:

General comment:

Avoid using suffer or suffering when referring to people with dementia.

Abstract

The reference to nation dementia plans currently stands alone as a sentence that does not link to the rest of the abstract and so the authors might consider removing this detail or developing it to improve the flow of the abstract.

Introduction

The structure of the introduction needs attention.

In paragraph two, page 1 lines 38 - 39 there is discussion of care homes which does not add to the paper, I would suggest removing this and including more discussion around people with dementia in hospitals and why dementia care pathways and dementia friendly environments are important. While there are sections in the introduction related to these, they are not explicit about why they are needed for people with dementia.

Page 1/2 lines 44-46 - the logic of this argument is unclear and I do not think that the reference supports the authors' interpretation.

Page 2, lines 62-63 there needs to be a reference to support this statement. It would also be helpful to the reader if there was an earlier, brief discussion of national dementia plans and the countries that have them or don't have them and why this is of relevance to dementia care in hospital settings.

Page 2 67 - 69 This sentence needs some attention, "These interventions have contributed to tackle the stigma of health organizations as unknown and dangerous environment." I don't think that stigma is the right word if it is referring to people with dementia experience of accessing health care. I also think that dangerous is the wrong word to be using. 

Methods

It would be helpful to set out clearly the aim of the review, not just the aim of the project as a whole or the aim of scoping reviews in general. The research question is included but could be more obvious perhaps by setting it on a different line with text "The scoping review aimed to answer the following research question:

[Research question]"

There needs to be more detail about the screening process, for example how was screening conducting and with how many researchers. 

There is no detail about data extraction or analysis of results, this needs to be included.

Results

Dementia care pathway. Table 1

Looking at some of the characteristics of some of the included papers, I was unclear how some papers met the inclusion criteria of being hospital based studies, for example Filippini et al 2004 there is mention of GP education, Ghiotti 2009 appears to be a community based pathway to avoid hospital admission. Their inclusion could be clarified with and additional column that, for example, states the setting.

Page 12 lines 172 to 182. Some of this text reads as if it is setting out the background rather than reporting the results. A few changes to the wording would address this, such as us of Authors suggest..., Authors found...

Some write up of the results should give a little more detail into the types of study when reporting results; e.g. a qualitative study of x found, a survey of [type of participants] reported...

Table 2

I am unclear about the inclusion of some papers, e.g. de Boer et al 2017 which is based in nursing homes not hospitals, Bossen 2010 which appears to be outdoor areas but is not clear if it relates to hospitals, Brawley 2007 which does not appear to be related to hospitals, Iaboni et al 2020 which I am unclear if this includes hospitals, Lillo-Crespo 2019 which does not appear to be hospital based. It may be that the inclusion of these papers just requires additional clarity in the table. 

Discussion

Page 20 lines 286 - 295

There needs to be more description about what prudent healthcare is and how it links to results reported in this manuscript.

Page 20 lines 296 - 314

This paragraph contains a lot of different concepts. It would be worth splitting the paragraph and developing some of the ideas further. For example, the authors highlight the importance of caregivers involvement during hospital admissions in hispanic-latin cultures and it would be worth developing this further.

Conclusion

The conclusion could make more reference to the findings from the review. 

Author Response

Dear Reviewer,

I highly appreciate your comments and suggestions to this paper. I believe it has really improved the manuscript. Please find below the answer to your comments:

Avoid using suffer or suffering when referring to people with dementia.Perfect, It has been changed.

Abstract

The reference to nation dementia plans currently stands alone as a sentence that does not link to the rest of the abstract and so the authors might consider removing this detail or developing it to improve the flow of the abstract. It has been removed.

Introduction

The structure of the introduction needs attention. It has been changed.

In paragraph two, page 1 lines 38 - 39 there is discussion of care homes which does not add to the paper, I would suggest removing this and including more discussion around people with dementia in hospitals and why dementia care pathways and dementia friendly environments are important. While there are sections in the introduction related to these, they are not explicit about why they are needed for people with dementia. Information has been included to reinforce the importance of care pathways for PwD.

Page 1/2 lines 44-46 - the logic of this argument is unclear and I do not think that the reference supports the authors' interpretation. It has been removed.

Page 2, lines 62-63 there needs to be a reference to support this statement. It would also be helpful to the reader if there was an earlier, brief discussion of national dementia plans and the countries that have them or don't have them and why this is of relevance to dementia care in hospital settings. Information has been included in order to support the statement and a table to show dementia national plans.

Page 2 67 - 69 This sentence needs some attention, "These interventions have contributed to tackle the stigma of health organizations as unknown and dangerous environment." I don't think that stigma is the right word if it is referring to people with dementia experience of accessing health care. I also think that dangerous is the wrong word to be using. It has been corrected.

Methods

It would be helpful to set out clearly the aim of the review, not just the aim of the project as a whole or the aim of scoping reviews in general. The research question is included but could be more obvious perhaps by setting it on a different line with text "The scoping review aimed to answer the following research question: It has been clarified in methods section.

[Research question]"

There needs to be more detail about the screening process, for example how was screening conducting and with how many researchers. There is a new section describing this requirement called “Data extraction and analysis”.

There is no detail about data extraction or analysis of results, this needs to be included. There is a new section describing this requirement.

Results

Dementia care pathway. Table 1

Looking at some of the characteristics of some of the included papers, I was unclear how some papers met the inclusion criteria of being hospital based studies, for example Filippini et al 2004 there is mention of GP education, Ghiotti 2009 appears to be a community based pathway to avoid hospital admission. Their inclusion could be clarified with and additional column that, for example, states the setting. We have clarified it in the discusison section explaining that these studies enrich the search as they are related with healthcare training and community based settings, please let us know if it is enough.

Page 12 lines 172 to 182. Some of this text reads as if it is setting out the background rather than reporting the results. A few changes to the wording would address this, such as us of Authors suggest..., Authors found...It has been included.

Some write up of the results should give a little more detail into the types of study when reporting results; e.g. a qualitative study of x found, a survey of [type of participants] reported.... Authors have include more information about the studies in the results description.

Table 2

I am unclear about the inclusion of some papers, e.g. de Boer et al 2017 which is based in nursing homes not hospitals, Bossen 2010 which appears to be outdoor areas but is not clear if it relates to hospitals, Brawley 2007 which does not appear to be related to hospitals, Iaboni et al 2020 which I am unclear if this includes hospitals, Lillo-Crespo 2019 which does not appear to be hospital based. It may be that the inclusion of these papers just requires additional clarity in the table. We have clarified it in the discusison section explaining that these studies enrich the search as they are related with healthcare environments and/or PwD needs from different perspectives that it could be helpful to find for readers considering their healthcare facilities.

Discussion

Page 20 lines 286 - 295

There needs to be more description about what prudent healthcare is and how it links to results reported in this manuscript. It has been explained in the 4th paragraph of discussion section.

Page 20 lines 296 - 314

This paragraph contains a lot of different concepts. It would be worth splitting the paragraph and developing some of the ideas further. For example, the authors highlight the importance of caregivers involvement during hospital admissions in hispanic-latin cultures and it would be worth developing this further. It has been explained in depth and paragraphs have been divided.

Conclusion

The conclusion could make more reference to the findings from the review. More information has been included referring to findings in conclusion section.

Thank you, 

Round 2

Reviewer 2 Report

Comment 1: There were still grammatical and spelling errors in the manuscript. I still have concerns about the quality of the writing. 

Comment 2: There were still abbreviations that had not been explained when first used in the main text. 

Comments 3 & 4  are addressed. 

Comment 5: Was the PRISMA-ScR form submitted? It is not attached to the manuscript. 

Comment 6: They have not considered or discussed the limitations of this review. For example, the lack of a quality assessment or formal synthesis. 

Reviewer 3 Report

I would like to thank the authors for making the amendments to the manuscript, I have no further comments. I would like to wish them all the best on their submission.